# RNAi-Mediated Screen of Primary AML Cells Nominates MDM4 as a Therapeutic Target in NK-AML with *DNMT3A* Mutations

**DOI:** 10.3390/cells11050854

**Published:** 2022-03-02

**Authors:** Olga Alexandra Sidorova, Shady Sayed, Maciej Paszkowski-Rogacz, Michael Seifert, Aylin Camgöz, Ingo Roeder, Martin Bornhäuser, Christian Thiede, Frank Buchholz

**Affiliations:** 1Medical Systems Biology, Faculty of Medicine, Technische Universität Dresden, 01307 Dresden, Germany; olga.sidorova@charite.de (O.A.S.); shady.sayed@tu-dresden.de (S.S.); maciej.paszkowski-rogacz@tu-dresden.de (M.P.-R.); 2Institute for Medical Informatics and Biometry (IMB), Technische Universität Dresden, 01307 Dresden, Germany; michael.seifert@tu-dresden.de (M.S.); ingo.roeder@tu-dresden.de (I.R.); 3Hopp Children’s Cancer Center Heidelberg, 69120 Heidelberg, Germany; aylin.camgoz@kitz-heidelberg.de; 4German Cancer Research Center (DKFZ), 69120 Heidelberg, Germany; martin.bornhaeuser@uniklinikum-dresden.de (M.B.); christian.thiede@uniklinikum-dresden.de (C.T.); 5National Center for Tumor Diseases (NCT/UCC), 01307 Dresden, Germany; 6Faculty of Medicine, University Hospital Carl Gustav Carus, Technische Universität Dresden, 01307 Dresden, Germany; 7Helmholtz-Zentrum Dresden—Rossendorf (HZDR), 01328 Dresden, Germany; 8Medical Clinic and Polyclinic I, University Hospital Carl Gustav Carus, Technische Universität Dresden, 01307 Dresden, Germany; 9German Cancer Consortium (DKTK), Partner Site Dresden, 01307 Dresden, Germany

**Keywords:** acute myeloid leukemia, DNMT3A, MDM4, RNAi, functional screen

## Abstract

*DNA-methyltransferase 3A* (*DNMT3A*) mutations belong to the most frequent genetic aberrations found in adult acute myeloid leukemia (AML). Recent evidence suggests that these mutations arise early in leukemogenesis, marking leukemic progenitors and stem cells, and persist through consolidation chemotherapy, providing a pool for AML relapse. Currently, there are no therapeutic approaches directed specifically against this cell population. To unravel therapeutically actionable targets in mutant *DNMT3A*-driven AML cells, we have performed a focused RNAi screen in a panel of 30 primary AML samples, all carrying a *DNMT3A* R882 mutation. As one of the strongest hits, we identified *MDM4* as a gene essential for proliferation of primary *DNMT3A^WT/R882X^* AML cells. We analyzed a publicly available RNA-Seq dataset of primary normal karyotype (NK) AML samples and found a trend towards MDM4 transcript overexpression particularly in *DNMT3A*-mutant samples. Moreover, we found that the MDM2/4 inhibitor ALRN-6924 impairs growth of *DNMT3A^WT/R882X^* primary cells in vitro by inducing cell cycle arrest through upregulation of p53 target genes. Our results suggest that MDM4 inhibition is a potential target in NK-AML patients bearing *DNMT3A* R882X mutations.

## 1. Introduction

Acute myeloid leukemia (AML) is a malignant cancer of the hematopoietic system. It is characterized by uncontrolled proliferation of immature blood cells, called leukemic blasts, caused by the impairment of differentiation programs in hematopoietic stem and progenitor cells (HSPCs). Conventional chemotherapy treatment can efficiently eliminate the leukemic blasts; however, it typically fails to eradicate the immature, quiescent populations of leukemic progenitors [1,2] These can drive leukemogenesis anew, creating a risk of disease recurrence (relapse), often with a more aggressive phenotype and a worse prognosis [3]. Thus, scientific efforts have been directed towards the development of targeted therapies that would eliminate these immature cell populations and therefore prevent recurrence of the disease in patients in remission. 

Unlike leukemic blasts that harbor several driver mutations, which can potentially be exploited as therapeutic targets (e.g., FLT3-ITD [4]), leukemic progenitors often lack therapeutically actionable mutated targets. A prominent example of this is normal karyotype AML (NK-AML) with mutations in *DNMT3A*. Recent studies in primary cells from AML patients have shown that the *DNMT3A* R882 mutation marks immature cells that are resistant to chemotherapy and represent a reservoir for AML relapse [5]. Remarkably, AML patients with *DNMT3A* mutations have shorter survival than *DNMT3A^WT^* patients, and this effect seems to be independent of the characteristic late co-mutations in *FLT3* or *NPM1 [6].* The mutant DNMT3A^R882X^ protein induces epigenetic changes in HSCs that distort the balance between self-renewal and differentiation [7,8], however detailed knowledge about the downstream targets of these changes remains scarce [9]. Thus, there is a clinical need for discovering vulnerabilities of this cell population that can be exploited as a therapeutic approach.

High-throughput genetic screens are a well-established technique for revealing genotype–phenotype connections. Genetic screens in cancer cell lines have been used to identify cancer vulnerabilities and in recent years very comprehensive databases of such vulnerabilities have been generated [10,11,12,13]. Nevertheless, confirming these dependencies in patient-derived material remains a challenging task, exposing the drawbacks of cell lines as AML models. While AML cell lines largely reflect the cytogenetics of the original malignancy [14], the transcriptional and epigenetic profiles of cancer cell lines may differ dramatically from those of primary cancer cells [15,16,17,18]. Previous work from our lab has shown that high-throughput genetic screens in primary AML cells are feasible and a powerful tool to nominate patient-specific targets [19]. Here, we describe a high-throughput RNAi-based screening approach to functionally profile primary *DNMT3A^WT/R882X^* cells from AML patients.

## 2. Materials and Methods

### 2.1. Cell Culture

The leukemic cell line OCI-AML3 was maintained in Alpha-MEM (Biochrom AG, Berlin, Germany) supplemented with L-Glutamine (2 mM; Invitrogen^TM^ Thermo Fisher Scientific, Waltham, MA, USA), Penicillin/Streptomycin (50 U/mL; Invitrogen^TM^) and 20% heat-inactivated FBS (Gibco^TM^ Thermo Fisher Scientific, Waltham, MA, USA). The leukemic cell line SET-2 was maintained in RPMI (Gibco^TM^ Thermo Fisher Scientific, Waltham, MA, USA) supplemented with Penicillin/Streptomycin (50 U/mL) and 20% heat-inactivated FBS. HEK293T and HeLa cells were cultured in DMEM (Gibco^TM^ Thermo Fisher Scientific, Waltham, MA, USA) supplemented with Penicillin/Streptomycin (50 U/mL) and 10% FBS. All cell lines were kept at 37 °C and 5% CO_2_ and were split twice a week. Primary AML cells were cultured in StemSpan Serum-Free Expansion Medium II (STEMCELL Technologies, Vancouver, BC, Canada) containing L-Glutamine (2 mM), Penicillin/Streptomycin (50 U/mL), the cytokines Flt3-Ligand (50 ng/mL), TPO (50 ng/mL), SCF (100 ng/mL) and IL-3 (20 ng/mL) (all from PeproTech Inc., Cranbury, NJ, USA) and the compounds StemRegenin1 (500 nM) and UM729 (50 nM) (both from STEMCELL Technologies, Vancouver, BC, Canada). Whenever possible, primary AML cells were maintained at high density (5−10 × 10^6^ cells/mL) and the medium was changed every second day.

### 2.2. DNMT3A R882H Mutation Correction

The lentiviral ABE8e-GFP-Puromycin construct was generated in our lab by replacing the cytidine base editor (CBE) in the plasmid pLenti-FNLS-P2A-GFP-PGK-Puro, a gift from Lukas Dow (Addgene plasmid # 110869; http://n2t.net/addgene:110869; RRID:Addgene_110869), with the ABE8e sequence, a gift from David Liu (Addgene plasmid # 138491; http://n2t.net/addgene:138491; RRID:Addgene_138491). The lentiviral particles were produced according to the protocol described in Section 2.9. The leukemic cell line SET2, carrying the *DNMT3A* R882H mutation, was transduced as described in the Section 2.10. One week after transduction, GFP-positive cells were sorted and afterwards constantly maintained in 2 ug/mL Puromycin. This stable SET2-ABE8e cell line was subjected to transduction with lentiviral particles containing pLenti-U6-gDNMT3A-EF-tdTomato-P2A-BlasR (see Section 2.8). The correction of the *DNMT3A* R882H mutation was verified by Sanger sequencing of the targeted *DNMT3A* locus, obtained through PCR amplification of the region from genomic DNA, isolated from the infected cell pool 5 days post transduction. The electropherograms were submitted to the EditR software [20] to evaluate the editing efficiency. The infected cell pool was maintained in culture and the fraction of the targeted cell population was measured every 4 days by analyzing the tdTomato signal on MACSQuant^®^ VYB (Miltenyi Biotec, Bergisch Gladbach, Germany). 

### 2.3. Thawing of Primary AML Cells

Frozen vials with primary AML cells were rapidly thawed in a 37 °C waterbath. The thawed cell suspension was added dropwise to 10 mL of the RPMI-1640 medium (Gibco^TM^ Thermo Fisher Scientific, Waltham, MA, USA) containing heparin (20 U/mL; Biochrom AG, Berlin, Germany), DNAse I (8 U/mL; Sigma-Aldrich^®^, St. Louis, MO, USA) and MgCl_2_ (4 mM; home-made) and incubated at 37 °C for 1 h. Next, cells were pelleted for 10 min at 300× *g* and resuspended in the culture medium. The cells were incubated overnight to allow cytokine-mediated activation and expansion before proceeding to B/T cell depletion. The cell viability was measured via trypan blue exclusion assay right after thawing as well as after the overnight cytokine induction. 

### 2.4. B/T Cell Depletion

After overnight cytokine induction, B and T cells were removed from the samples using the EasySep^TM^ Human CD19/CD3 Positive Selection Kit II (STEMCELL Technologies, Vancouver, BC, Canada) according to manufacturer’s instructions with the difference that the supernatant was kept and the B and T cells containing tubes were discarded. Cells were counted and a sample of 1 × 10^5^ cells was used for flow cytometry to check for successful depletion of the lymphocytes. In brief, cells were washed with PBS-EB (PBS with 2.5 mM EDTA and 1% BSA) and incubated with anti-human CD2-FITC and anti-human CD19-PE-Cy7 (both from Biolegend, San Diego, CA, USA) antibodies for 30 min at 4 °C in the dark. The stained cells were washed twice in PBS-EB and resuspended in PBS containing 1 μg/mL DAPI (Sigma-Aldrich, St. Louis, MO, USA). Flow cytometry was performed on the BD LSR II (Becton Dickinson, Franklin Lakes, NJ, USA) machine and the data analyzed using FlowJo software (FlowJo, LLC, Becton Dickinson, Franklin Lakes, NJ, USA).

### 2.5. Transduction of Primary AML Cells with the shRNA Library

The customized focused shRNA library was described previously [21]. The packaging of the shRNA library into lentiviral particles was performed by Sirion-Biotech GmbH (Planegg, Germany). The viral preparation had a functional titer of 4.2 × 10^9^ IU/mL as detected by the manufacturer in HEK293 cells. B and T cell-depleted primary AML cells were counted and 1 × 10^7^ (or the maximum available number) were infected with the shRNA library according to the spinoculation protocol (see Materials and Methods Section 2.8) at the multiplicity of infection (MOI) ranging from 2 to 10, depending on the available cell number. Note that because primary AML cells are difficult to transduce, this MOI corresponds to an infection rate of 10–50%, depending on the primary sample, not exceeding the integration rate of one copy of the expression plasmid per cell. After 20 h of incubation at 37 °C and 5% CO_2_ the cells were washed twice in PBS, resuspended in the culture medium and returned to the cell culture incubator. At 96 h post transduction, the infection rate was analyzed by assessing the percentage of RFP-positive cells using BD Canto II (Becton Dickinson, Franklin Lakes, NJ, USA) and the FlowJo software. The cells were counted, and half of the cells were collected as a baseline sample and frozen away at −20 °C until genomic DNA isolation. The remaining cells were cultured for 20 more days and then collected as a final timepoint sample. 

### 2.6. Preparation of the Samples for Deep Sequencing

Genomic DNA was isolated from primary AML cells carrying the shRNA library with the QIAamp^®^ DNA Blood Mini Kit (Qiagen, Hilden, Germany) according to the manufacturer’s instructions. Two rounds of PCR were performed using Titanium^®^ Taq PCR Kit (Takara Clontech, Kusatsu, Japan) to amplify the library barcodes (primers #4 and #5, Appendix A; with up to 50 μg DNA in a 100 μL reaction volume) and to attach the adapters for Illumina sequencing (primers #6 and #7, Appendix A; 1 μL of a 1:500 dilution of PCR1 as input). The PCR products were purified with the ISOLATE II PCR and Gel Kit (Bioline Meridian Bioscience Inc., Cincinnati, OH, USA) according to the manufacturer’s instructions, quantified using Qubit Fluorometer (Invitrogen™, Thermo Fischer Scientific, Waltham, MA, USA) and submitted to the Deep Sequencing Facility (CRTD, Dresden, Germany), where multiplexed libraries were prepared. The sequencing was performed with the NextSeq 500 sequencer (Illumina Inc., San Diego, CA, USA) at the sequencing depth of 1 × 10^7^ reads per sample.

### 2.7. Deep Sequencing Data Analysis

For each patient sample, the baseline (day 0) and the final (day 20) timepoints were sequenced. The raw read counts were transformed into counts per million (CPM), normalized using the cyclic loess method and grouped by gene (on average 12 shRNAs/gene). The Welch’s rank sum *t*-Test [22] was used to compare the normalized CPM between day 20 and day 0 timepoints and calculate a p-value for each gene, which was corrected for multiple testing by computing FDR-adjusted *p*-values and Q-values based on Storey’s method [23]. Finally, a ratio of Day 20 to Day 0 normalized CPM (referred to as fold change or FC) was then built, log-transformed and used to generate a volcano plot. An shRNA species, i.e., all shRNAs targeting the same gene, was regarded as significantly enriched or depleted when the corresponding Q-value was <0.05. As a measure of technical screen success, the abundance of the shRNAs directed against control genes *PSMA1*, *PSMA3*, *RPS13*, *RPL6* and *RPL30* was evaluated. A screen was regarded as successful if at least one of the control genes was significantly depleted (Q-value < 0.05, FC < −0.2). After the selection of successful screens, the corresponding data were re-analyzed starting from the raw read count. The analysis pipeline was the same as described above, with the difference that the grouping per gene was performed across all successful samples, instead of per patient. This step was implemented to increase the statistical power of the analysis and the probability to identify common trends within the analyzed group of samples. Genes were nominated as candidate dependencies when the depletion criteria (Q-value < 0.05 and FC < −0.2) were met. 

### 2.8. Cloning of shRNAs and gRNAs

The gRNA for correction of the *DNMT3A* R882H mutation in SET-2 cells using the adenine base editor ABE8e was designed manually. For validation of the screen results with RNAi, the two top scoring shRNAs per candidate gene, according to the obtained screen data, were selected (Appendix A). For validation of the screen results with the CRISPR system, GPP sgRNA Designer [24], the CHOPCHOP V3 [25] and InDelphi [26] tools were used. All oligos were synthesized by metabion international AG (Planegg, Germany). The oligos were processed and cloned into pLenti-U6-tdTomato-P2A-BlasR (LRT2B) (a gift from Lukas Dow, Addgene #110854), pRSI12.U6-shRNA.UbiC-GFP-T2A-Puromycin (Cellecta) or pL.CRISPR.EFS.GFP (a gift from Benjamin Ebert, Addgene plasmid # 57818), respectively, following the protocol described in [27]. In brief, the oligos were phosphorylated and annealed with T4 PNK in T4 ligation buffer, digested with BbsI and ligated with T4 ligase in Tango buffer (all from New England Biolabs, Ipswich, MA, USA). The ligate was transformed into home-made electrocompetent XL1 blue cells at 1700 V. Home-made SOC medium was added, and the bacteria were plated on Ampicillin agar plates. The colonies were picked after overnight incubation at 37 °C and a colony PCR was performed with MyTaq Red Polymerase (Bioline Meridian Bioscience Inc., Cincinnati, OH, USA) following the manufacturer’s instructions using the primers U6-seq-F and shRNA-seq-R (for shRNA inserts) or U6-seq-F and gRNA-seq-R (for gRNA inserts) (Appendix A) with the indicated annealing temperatures, an extension time of 20 s and 30 cycles. The PCR products were sent to the sequencing service provider Microsynth Seqlab GmbH (Göttingen, Germany). The positive clones were grown overnight in 230 mL LB under Ampicillin selection (50 μg/mL) and the cultures were processed with Qiagen Plasmid Maxi Kit (Qiagen, Hilden, Germany) according to manufacturer’s instructions. The obtained transfer plasmids were used for virus production (see Section 2.9 Virus production).

### 2.9. Virus Production

Fifteen-million HEK293T cells were seeded per one T-175 flask and transfected on the next day at 80% confluency with 2 µg pMD2.G (Addgene plasmid #12259), 6 µg psPAX2 (Addgene plasmid #12260) and 10 µg of the transfer vector using PEI (1 mg/mL), according to the standard protocol. After 20 h, the medium was changed to complete DMEM, and at 72 h post transfection the viral supernatant was collected, passed through 0.45 μm filter and centrifuged for 1.5 h at 100,000× *g* and 4 °C. The supernatant was decanted, and the viral pellets resuspended in PBS for at least 4 h at 4 °C on a rocking table. Where indicated, the lentiviral particles were concentrated using Amicon Ultra-15 Centrifugal Filter Devices (Merck Millipore, Burlington, MA, USA). For long-term storage the virus particles were kept in cryovials at −80 °C. 

### 2.10. Transduction of Cells by Spinoculation

Prior to transduction, microplates were coated with RetroNectin^®^ (Takara Clontech, Kusatsu, Japan), according to manufacturer’s instructions. Cells were infected in the presence of protamine sulfate (Merck Millipore, Burlington, MA, United States; 50 µg/mL final concentration) at 880× *g* and 37 °C for 45 min and then incubated for 20 h until the medium was changed. To introduce the shRNA library into primary AML cells, 50 μL of virus particles was used in a total volume of 4 mL in a 6-well plate (see Materials and Methods Section 2.4). To introduce individual shRNAs into cell lines, 200 μL of concentrated virus particles was used in a total volume of 1 mL in a 24-well plate (see Materials and Methods Section 2.9). To introduce a gRNA and Cas9 into cell lines, 200 μL of concentrated virus particles was used in a total volume of 400 μL in a 48-well plate (see Materials and Methods Section 2.10).

### 2.11. Knockdown of Candidate Genes in Cell Lines with RNAi

In a 24-well plate, 1 × 10^6^ cells were infected with 200 μL Amicon-concentrated lentiviral particles, by spinoculation to stably express an shRNA and GFP. The next day, the cell pool was separated into two fractions. The first fraction of 1.5 × 10^5^ cells was split into three wells of a U-bottom 96-well plate and maintained in culture for flow cytometric analysis. The cells were analyzed for GFP expression on MACSQuant^®^ X (Miltenyi Biotec, Bergisch Gladbach, Germany) every second day starting at 48 h until 9 days after transduction. The second pool was transferred into a 6-well plate, where the cells were selected with puromycin (Sigma-Aldrich, St. Louis, MO, USA; 2 μg/mL final concentration) for two days starting at 48 h after transduction. The selected cells were collected for RNA isolation and qPCR analysis of the knock down efficiency.

### 2.12. Knock out of Candidate Genes in Cell Lines with CRISPR/Cas

In a 48-well plate 1 × 10^5^ cells were infected with 200 μL Amicon-concentrated lentiviral particles by spinoculation to stably express a gRNA and Cas9-GFP. The next day the cells were split into three wells of a U-bottom 96-well plate and maintained for flow cytometric analysis. The cells were analyzed for GFP expression on MACSQuant^®^ X (Miltenyi Biotec, Bergisch Gladbach, Germany) every second day starting at 96 h until 14 days after transduction.

### 2.13. Quantitative Real-Time PCR

RNA was isolated from the cells using RNeasy Mini Kit and RNase-Free DNase Set (Qiagen, Hilden, Germany) according to manufacturer’s instructions. Reverse transcription of 1 μg total RNA was performed with oligo(dT)20 (50 μM) using SuperScript IV Reverse Transcriptase kit (Invitrogen™, Thermo Fischer Scientific, Waltham, MA, USA). The qPCR was done with the ABsolute QPCR Mix with ROX (Invitrogen™, Thermo Fischer Scientific, Waltham, USA) in CFX96 Touch Real-Time PCR Detection System (Bio-Rad laboratories, Hercules, CA, USA). The primer list can be found in the Appendix A. The primer sequences for the amplification of GADD45a, PUMA and BAX were taken from [28].

### 2.14. Western Blot

HeLa cells transduced with pRSI12.U6-shRNA.UbiC-GFP-T2A-Puromycin were collected, washed in PBS, counted and resuspended in Cell Lysis Buffer (Cell Signaling Technology, Danvers, MA, USA) supplemented with Protease Inhibitor Cocktail (Cell Signaling Technology, Danvers, MA, USA) at 10 μL per 1 × 10^5^ cells. After 10 min incubation on ice, the suspension was spun down for 15 min at 14,000× *g* and the supernatant transferred into a new tube. The total protein concentration was determined using the BCA assay following the manufacturer’s instructions (Pierce™ BCA Protein Assay Kit, Thermo Scientific™, Waltham, MA, USA). The lysate containing 60 µg total protein was mixed with the 4x NuPAGE LDS Sample Buffer (Invitrogen™, Thermo Fischer Scientific, Waltham, MA, USA), boiled at 70 °C for 10 min and loaded onto NuPAGE 4–12% Bis-Tris Protein Gel (Invitrogen™, Thermo Fischer Scientific, Waltham, MA, USA). The gel was run at the constant voltage of 130 V. The transfer of the proteins onto Amersham Protran 0.45 NC nitrocellulose membrane (GE Healthcare, Chicago, IL, USA) was done with the semi-dry blotting method at 50 mA. The membrane was blocked with 5% milk in PBS-Tween (PBST, 0.1%) for 30 min at room temperature prior to overnight incubation with the primary anti-MDM4 and anti-GAPDH antibodies (Proteintech, Rosemont, IL, USA; Cat.No. 17914-1-AP, 500× dilution and Origene, Rockville, MD, USA; Cat.No. TA302944, 1000× dilution, respectively) at 4 °C, diluted in 5% milk in PBST. The membrane was washed 3 times for 10 min in PBST and incubated with secondary antibodies (IRDye 680LT Donkey anti-Goat IgG (H + L), Cat.No. 926-68024 and IRDye 800CW Donkey anti-Rabbit IgG (H + L), Cat. No. 926-32213), diluted in 5% milk in PBST (1:10,000), for 1 h at room temperature. After washing, the membrane was imaged with Odyssey^®^ CLx Imaging System (Li-cor Biosciences GmbH, Bad Homburg vor der Höhe, Germany).

### 2.15. ALRN-6924 Treatment and Cell Cycle Analysis

ALRN-6924 was a kind gift from Martin Bornhäuser (Medical Clinic and Polyclinic I, University Hospital Carl Gustav Carus, TU Dresden, Dresden, Germany). The stock solution of 10 mM was diluted in 10% DMSO (Merck, Darmstadt, Germany) in water to obtain predilutions. The final dilutions were made in the primary AML culturing medium, to a final concentration of DMSO of 0.01%. For the cell cycle analysis, primary AML cells were thawed, purified from B/T cells as described above and cultured in the medium with 5 µM ALRN-6924 for 24 h. The staining for cell cycle analysis was performed with the Click-iT™ EdU Alexa Fluor™ 647 Flow Cytometry Assay Kit (Invitrogen™, ThermoScientific, Waltham, MA, USA) according to the manufacturer’s instructions. The flow cytometry analysis was carried out on MACSQuant^®^ X (Miltenyi Biotec, Bergisch Gladbach, Germany). 

## 3. Results

### 3.1. The DNMT3A R882H Mutation Is Not Essential in the Leukemic Cell Line SET2

Because *DNMT3A* mutations are an early oncogenic event and *DNMT3A*-mutant cells frequently resist chemotherapy, we first wanted to investigate whether mutant *DNMT3A*-driven leukemic cells depend on this mutation for survival and proliferation. To this end, we have utilized the recently described adenine base editing system (ABE8e) [29] to correct the *DNMT3A* R882H mutation in the leukemic cell line SET2. We generated a transgenic SET2 line by transducing the cells with a lentiviral ABE8e-GFP-Puromycin construct. GFP-positive cells after puromycin selection were then infected with a lentiviral construct to deliver gRNAs to correct the *DNMT3A* R882H mutation. Five days post transduction of a lentiviral tdTomato reporter carrying the gRNA against the *DNMT3A^R882H^* locus into the transgenic SET2-ABE8e-GFP-Puromycin cells, we observed a 14% A to G correction rate in the targeted region (Figure 1a), demonstrating that the base-editor is effective in reverting the *DNMT3A* mutation. We then followed the cells and monitored the percentage of tdTomato-positive cells over time. We did not observe a respective drop in the tdTomato signal over time (Figure 1b), indicating that the cells with corrected *DNMT3A* can survive and proliferate normally. This experiment suggests that SET2 cells do not depend on the mutant DNMT3A function for survival and proliferation.

### 3.2. RNAi Screening in Mutant DNMT3A-Driven Primary AML Cells

To investigate other potential dependencies of mutant *DNMT3A*-driven leukemic cells, we decided to perform RNAi-based functional screens on primary AML cells. The primary bone marrow samples selected for screening all harbored a *DNMT3A* p.R882 mutation. All the samples were additionally carrying a frameshift mutation in *NPM1* (p.W288Xfs*12; also called NPM1c). The mutational information of the 30 samples selected for the screening is summarized in the Appendix A. The screen procedure is illustrated in Figure 2a.

Because the availability of the patient samples was limited and did not allow for the isolation of a sufficient number of *DNMT3A^WT/R882X^* progenitor cells, the screens were performed on the bulk mononuclear cells. Based on the available cell number and the screen coverage requirements, we opted for a focused library. The rationale for the library composition was the known role of epigenetic regulators during early steps of leukemogenesis [30]. Thus, we aimed to probe the dependency of *DNMT3A^WT/R882X^* AML cells on epigenetic and transcriptional factors. The library consisted of ~6500 shRNAs targeting 540 genes (average 12 shRNA per gene), including one negative and five positive controls [21]. 

Unlike cancer cell lines, in vitro cultured primary cells are sensitive to manipulation procedures and have a limited proliferation capacity. Thus, functional screens in primary AML cells are a challenging task. Overall, 30 unique *DNMT3A^WT/R882X^* patient samples were screened. The total number of screened samples amounted to 39, considering replicate screens for some of the patient lines. From these, 34 samples, or 90%, survived the in vitro propagation, despite the initial drop of viability (Figure 2b, 42% median viability directly after thawing vs. 14% median viability 24 h after thawing; range at 24 h after thawing: 1–55%). In turn, 59% of these samples were proliferative in culture, i.e., the number of live cells increased between Day 0 and Day 20 of the screen (see Appendix A and Appendix A). As expected for primary material, the achieved library coverage was highly variable across samples (Figure 2c; 56× median representation at T_0_, range 10× to 300×). Compared to cell viability and the proliferation rate in culture, library coverage appeared to have the biggest impact on the behavior of positive controls (Appendix A), suggesting that the absolute number of shRNA-carrying viable cells is a major determinant of the screen success. Based on the performance of the shRNAs targeting the positive control genes *PSMA1*, *PSMA3, RPS13*, *RPL6* and *RPL30* compared to the non-targeting control shRNAs against luciferase (*LUC*), nine screens were considered successful. (Figure 2c and Appendix A).

To emphasize common trends and increase the statistical power of the analysis, we pooled the count data of the nine successful screens and compared the shRNA counts between Day 20 and day 0, using the Wilcoxon rank sum test. After this step, all positive control shRNAs were among the top-scoring hits, demonstrating that the obtained results appear valid (Figure 3a). Additionally, known common essential genes appeared as hits, such as the cohesin complex genes *SMC1a/2/3/4*. Based on this analysis, we selected nine candidate genes for validation experiments (Appendix A). To further shortlist candidates to study in primary cells, we used the leukemic cell line OCI-AML3, which genetically closely resembles the primary patient cohort (*DNMT3A^WT/p.R882C^, NPM1^WT/p.W288Xfs*12^*), but is easier to maintain and transduce. We tested the on-target activity of the two highest-scoring shRNAs from the screening library per gene of interest in a qPCR and a GFP reporter assay. An shRNAs was considered validated, when the reduction of mRNA levels upon the knock down was >50% and the depletion of the GFP reporter signal was >50% by day 9 post transduction. The qPCR revealed that the knock down was on target in 15 out of 18 tested shRNAs (Appendix A). In the GFP reporter assay, a consistent depletion phenotype was achieved with four hits—DBF4 zinc finger (DBF4), Glucocorticoid Modulatory Element Binding protein 1 (GMEB1), Thioredoxin Domain Containing Protein 9 (TXNDC9) and Mouse Double Minute 4 (MDM4) (Appendix A). We decided to focus on MDM4, because it is a well characterized protein with a prominent role in cell cycle regulation and has potential for chemical inhibition.

In summary, our results show that cell viability is the bottleneck of in vitro functional screens in primary AML cells. Nevertheless, depletion of positive controls can be used to nominate screening samples that were successful. Additionally, we could demonstrate that, in the absence of biological replicates, pooling of the count data obtained in different patient lines emphasized the common trends and allowed for nominating potential common dependencies of the analyzed sample group.

### 3.3. Loss of MDM2/4 Activity Stops the Proliferation of DNMT3A^WT/p.R882C^/NPM1^WT/p.W288Xfs*12^ Primary AML Cells

One of the strongest hits that emerged from the screen was Mouse Double Minute 4 (MDM4) (Figure 3a). MDM4 is a known regulator of the activity of the major tumor suppressor p53. Like its paralog Mouse Double Minute 2 (MDM2), it has an inhibitory effect on p53, preventing it from initiating a stress response program (reviewed in [31]). To confirm that *MDM4* was expressed in our primary samples, we analyzed RNA-Seq data available for our extended *DNMT3A^WT/R882X^* patient cohort. All the samples expressed *MDM4*, to a level comparable to that of the housekeeping genes *PSMD6, EIF4H* and *PSMA1 [32]* (Figure 3b). Next, we validated the phenotype of *MDM4* knockdown in two primary patient samples from the screened cohort, SAL1788 (*DNMT3A^WT/p.R882C^*, *NPM1^WT/WT^*, *FLT3^WT/WT^*) and SYT59 (*DNMT3A^WT/p.R882H^*, *NPM1^WT/p.W288Xfs*12^*, *FLT3^WT/WT^*), using a lentiviral GFP reporter system. The expression of either of the two shRNAs against *MDM4* led to a rapid depletion of primary AML cells in vitro. Of note, primary cells from different patients displayed unequal sensitivity to *MDM4* knock down, possibly reflecting the variable dependency of the original tumors, underlined by their genetic and epigenetic profiles (Figure 3c). We confirmed the on-target activity of the employed shRNAs on mRNA (Appendix A) and protein (Appendix A) levels. Furthermore, we utilized the CRISPR/Cas9 system to independently validate the dependency of *DNMT3A^R882X^*-driven leukemic cells on MDM4, using OCI-AML3 cells as a model (Appendix A). Continuous expression of Cas9 and a gRNA against *MDM4* led to the depletion of targeted cells over time (Appendix A), although the dynamics of the cell depletion here was slower than upon shRNA-mediated knockdown (shRNA vs. gRNA: depletion to 50% within 9 days vs. 14 days). To test the efficiency of the employed gRNA, we transduced HEK293T cells with a GFP-reporter construct expressing Cas9 and MDM4-gRNA and analyzed the targeted genomic locus by Sanger sequencing 5 days after infection. The sequencing revealed an in-frame:frameshift ratio of 1.86 (Appendix A). Thus, the slow dynamics of cell depletion can be explained by the low efficiency of the gRNA-mediated disruptive indel generation.

A recent report from Carvajal et al. demonstrated that MDM4 is particularly highly expressed in AML cell lines and patient-derived AML cells [28]. This study did not, however, investigate the potential differences between the cytogenetic subgroups of AML. Using a publicly available RNA-Seq dataset of AML samples (GSE67040), we compared the expression of MDM4 in *DNMT3A^WT/R882X^* and *DNMT3A^WT/WT^* normal karyotype (NK) AML. This analysis revealed a clear, albeit not statistically significant, trend towards higher MDM4 expression in *DNMT3A^WT/R882X^* samples (Figure 4). This trend could not, however, be confirmed in the normal karyotype AML samples from the TCGA cohort (data not shown).

Carvajal and colleagues further reported that the dual inhibition of MDM2/MDM4 with a novel inhibitor ALRN-6924 induced apoptosis and cell cycle arrest in leukemic cells accompanied by a survival advantage in xenotransplantation mouse models [28]. Of note, this study also provided evidence for the sensitivity of enriched leukemic progenitor populations to MDM2/4 inhibition and nominated MDM2/4 as therapeutically actionable targets for p53 wild type tumors [28]. Strikingly, Carvajal and colleagues observed that the leukemic cell line OCI-AML3 which, similar to our tested cohort, bears the mutations *DNMT3A^WT^/^p.R882C^* and *NPM1^WT/p.W288Xfs*12^*, had a high resistance to ALRN-6924 treatment compared to other leukemic cell lines, despite all having the p53 signaling pathway intact [28]. Although the study also reports the response of several primary AML samples to ALRN-6924 treatment, it does not indicate the mutational profiles of these cells. Thus, how distinct genetic subgroups of AML respond to ALRN-6924 treatment remains unknown. As our screen data suggested that primary *DNMT3A^WT/R882H^/ NPM1^WT/p.W288Xfs*12^* AML cells are sensitive to MDM4 depletion, we decided to investigate whether MDM2/4 inhibition is effective in these cells. We applied the dual chemical inhibitor ALRN-6924 described in [28] on two primary samples—SAL1788 (*DNMT3A^WT/p.R882C^*, *NPM1^WT/WT^*, *FLT3^WT/WT^*) and SYT50 (*DNMT3A^WT/p.R882H^*, *NPM1^WT/p.W288Xfs*12^*, *FLT3-ITD*).

The addition of ALRN-6924 to the culture medium reduced the counts of primary *DNMT3A^WT/R882H^/ NPM1^WT/p.W288Xfs*12^* AML cells in a time- and concentration-dependent manner (Figure 5a). This growth retardation was accompanied by a dramatic increase in expression of p53 target genes, particularly the cell cycle regulators p21 (mean fold change (fc) = 30.0) and GADD45a (mean (fc) = 22.9), as well as proapoptotic regulators PUMA (mean (fc) = 15.0) and BAX (mean (fc) = 3.3) (Figure 5b). MDM2 was found to be overexpressed as well (mean (fc) = 25.1), while MDM4 levels remained virtually unchanged (mean (fc) =1.5), in line with the previous evidence of the existence of an MDM2-p53 negative feedback loop [33,34]. An assay of the distribution of the primary AML cells across the cell cycle upon treatment with ALRN-6924 demonstrated a strong reduction of cells in S-phase with simultaneous accumulation of cells in the G0/1 phase, indicating cell cycle arrest (Figure 5c). Moreover, the induction of apoptosis suggested by the qPCR in the sample SYT1788 was confirmed through the increased fraction of cells in the Sub-G1 phase of the cell cycle upon treatment with ALRN-6924 (Appendix A). Thus, dual inhibition of MDM2/4 in primary *DNMT3A^WT/R882H^/NPM1^WT/p.W288Xfs*12^* AML cells caused cell cycle arrest through upregulation of p53 target genes, in particular *p21* and *GADD45a*, which resulted in growth retardation in vitro.

## 4. Discussion

Since the *DNMT3A* R882 mutation was found to mark early leukemic progenitors [5], efforts have been directed towards finding and exploiting vulnerabilities of this cell population for therapeutic applications. In this work, we have shown that mutant DNMT3A is not required for in vitro survival and proliferation of leukemic blasts by correcting the R882H mutation in SET2 cells with an adenine base editor. More subtle molecular changes resulting from the inactivation of this mutation need to be investigated in follow-up studies. Single-cell RNA-Seq/CHIP-Seq experiments could help delineate the role of DNMT3A in distinct cell populations. The response of early *DNMT3A*-mutant leukemic progenitors to the restoration of the wild type DNMT3A activity also remains to be investigated. Because the acquisition of *DNMT3A* mutations in HSCs leads to clonal hematopoiesis and often represents the first step of leukemogenesis [35,36,37,38,39], we speculate that reinstatement of the normal DNMT3A functions would be sufficient for the affected cell to return to its native state, i.e., revert the methylation state and the proliferation rate back to the wild type levels. Isolation of leukemic progenitor cells followed by the correction of the *DNMT3A* R882 mutation and a comparative analysis of methylation patterns and proliferative behavior in the corrected vs. non-corrected cells would be required to test this hypothesis.

*DNMT3A*-mutant clonal cell populations have been implicated in leukemogenesis initiation [5,40], chemotherapy resistance [5,30] and relapse [5,30,41]. Nevertheless, a targetable vulnerability of *DNMT3A*-mutant leukemic cells has not yet been found. We have functionally profiled primary *DNMT3A^WT/R882^* AML cells, aiming to identify cellular dependencies that can be exploited as therapeutic targets. To the best of our knowledge, such a large panel of primary human AML cells has not been subjected to functional screening before. Apart from the limited sample availability, such experiments are constrained by the viability and in vitro growth capacity of primary cells. Our data show that large cell numbers are required to account for the viability drop as the primary cells are adapting to in vitro culture conditions. Furthermore, some primary AML samples tend to rapidly differentiate in culture (data not shown), calling for improvements of the culture conditions to allow long term expansion. Methods such as co-culture with feeder cells [42,43,44] or in vivo propagation [45,46,47] have been shown to be superior to the in vitro culture with regard to the viability and proliferation capacity of primary cells. However, technical requirements, such as isolating of the population of interest in sufficient numbers/purity and ensuring high library coverage with large-scale screens, respectively, may be more challenging to meet with these methods. Using our screen setting (~6500 shRNAs), we have experimentally demonstrated that with the current culturing techniques a library representation of ~60× can be achieved and is sufficient to detect phenotypic effects (positive controls). If the primary material is scarce, we recommend to pool count data from multiple samples to emphasize common trends and identify shared hits, provided a rational pooling scheme (e.g., similar clinical and mutational profiles).

Our focused screen in primary *DNMT3A^WT/R882^* AML cells nominated several potential candidates, including the recently emerged promising target MDM4. A recent observation made by Carvajal and colleagues in the leukemic cell line OCI-AML3 (*DNMT3A^WT/p.R882C^*, *NPM1^WT/p.W288Xfs*12^*) suggested that these may be less responsive to MDM2/4 inhibition than other p53 WT cells [28]. In contrast, in this work we have shown that a knockdown of MDM4 or a chemical inhibition of MDM2/4 in primary *DNMT3A^WT/p.R882^*, *NPM1^WT/p.W288Xfs*12^* leukemic cells (SYT50 and SYT59) is sufficient to induce growth retardation in vitro and observed a similar effect in *DNMT3A^WT/p.R882^*, *NPM1^WT/WT^* cells (SAL1788). Moreover, we found a trend towards higher MDM4 expression in mutant *DNMT3A*-driven primary NK-AML cells, as compared to NK-AML without *DNMT3A* R882 mutations. Thus, our data suggest that MDM4 is an attractive antileukemic target in NK-AML subtypes with mutant DNMT3A.

## Figures and Tables

**Figure 1 cells-11-00854-f001:**
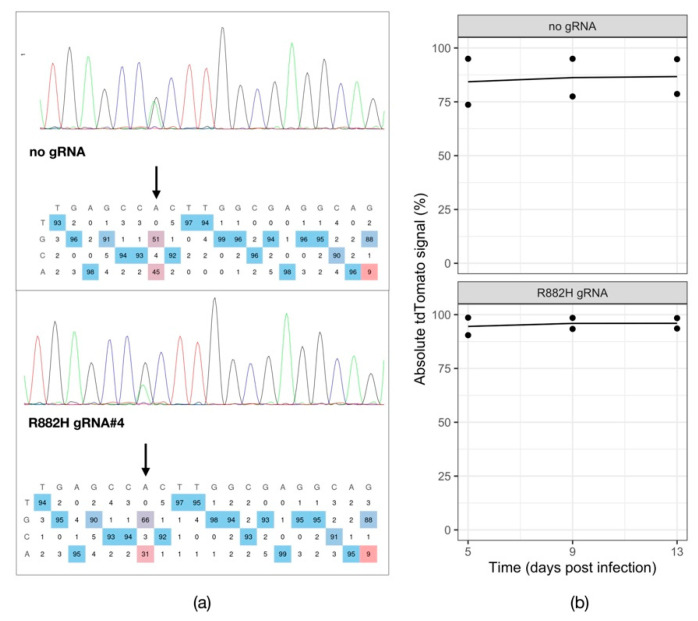
Testing the dependency of SET2 cells on mutant DNMT3A. (**a**) Chromatograms visualizing the Sanger sequencing reads of the PCR-amplified *DNMT3A* R882H genomic locus in SET2-ABE8e-GFP-Puro transgenic cell line expressing a tdTomato reporter alone (no gRNA) or in combination with a gRNA targeting the R882H mutation (R882H gRNA). (**b**) tdTomato reporter assay tracking the percentage of gRNA-expressing SET2-ABE8e-GFP-Puro cells over time (*n* = 2).

**Figure 2 cells-11-00854-f002:**
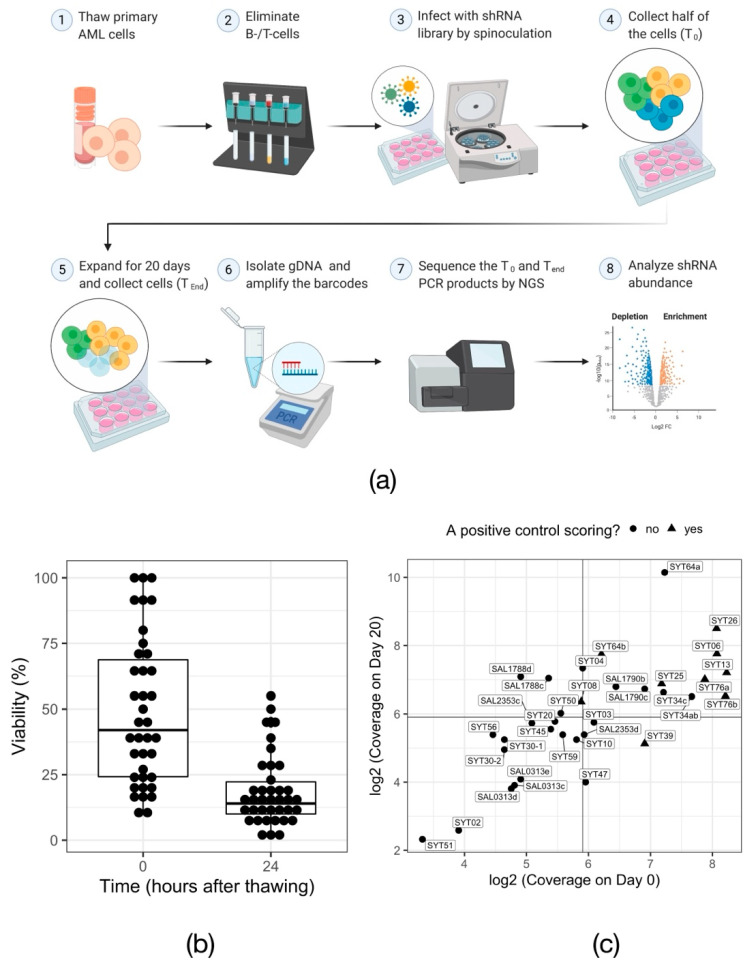
RNAi screen in primary AML samples. (**a**) Overview of the experimental procedure for RNAi-mediated in vitro screening of primary AML cells. Created with BioRender.com (**b**) Cell viability measured via trypan blue exclusion assay immediately after thawing of the samples (0 h) and after overnight cytokine induction (24 h). Each data point represents one primary AML sample. (**c**) Dot plot visualizing the shRNA representation, calculated based on the number of cells collected for gDNA isolation, in every primary sample at the beginning (Day 0) and the end (Day 20) of the screen. Solid lines mark the arbitrary threshold of 60× representation.

**Figure 3 cells-11-00854-f003:**
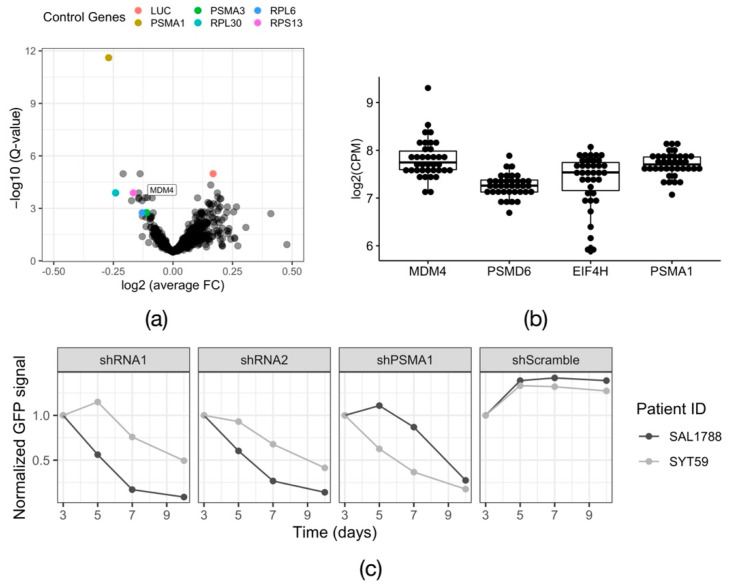
Validation of the screen hit MDM4. (**a**) Volcano plot visualizing the results of the combined RNAi screens in 9 primary patient samples. FC—fold change; LUC—luciferase. The *p*-values were calculated with the Wilcoxon rank-sum test and corrected for multiple testing using Storey’s method to obtain the Q-value. (**b**) Expression data for the candidate gene MDM4 and the housekeeping genes PSMD6, EIF4H, PSMA1 obtained from RNA-Seq of 37 *DNMT3A* R882X AML samples, including 18 samples in which the functional screens were performed. CPM—counts per million. (**c**) Growth dynamic of two primary *DNMT3A* R882X AML primary samples (SAL1788 and SYT59) expressing an shRNA against MDM4 (shRNA1 and shRNA2). The positive control PSMA1 (shPSMA1) and the non-targeting control (shScramble) are indicated. *n* = 1.

**Figure 4 cells-11-00854-f004:**
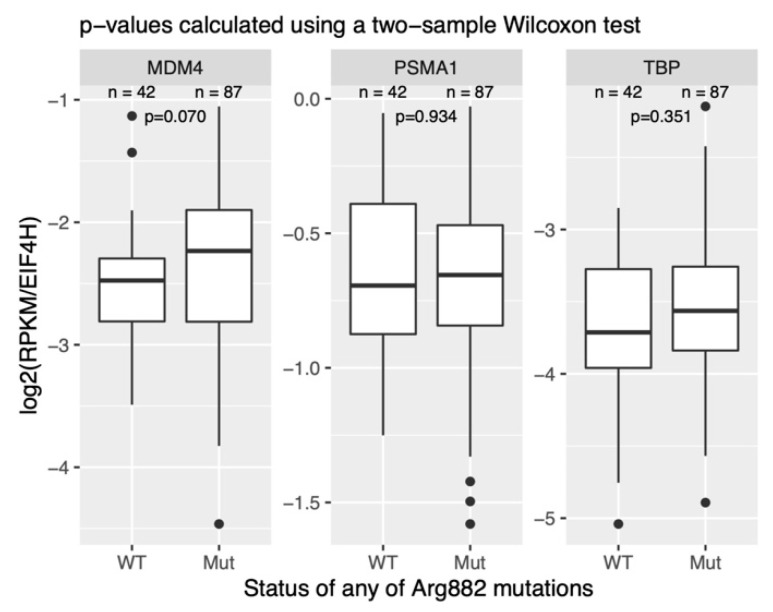
Comparison of MDM4 transcript expression in NK-AML samples with and without the *DNMT3A* R882H mutation. The expression values across samples are normalized to the housekeeping gene *EIF4H.* PSMA1 and PSMD6 are housekeeping controls. n is the number of samples in each group. The data was retrieved from the publicly available dataset GSE67040. RPKM—reads per kilobase of transcript per million of mapped reads.

**Figure 5 cells-11-00854-f005:**
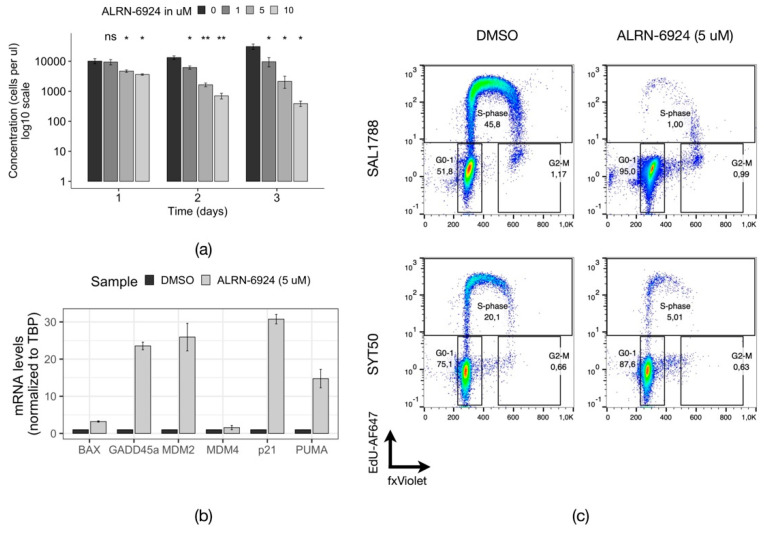
Effects of MDM2/4 inhibition in primary *DNMT3A^WT/R882^* AML cells. (**a**) Normalized counts of live primary AML cells (SAL1788) after the treatment with indicated concentrations of ALRN-6924 (*n* = 2). The samples were compared to the DMSO treated sample with a parametric two-sided Student’s t-test (*: *p* ≤ 0.05; **: *p* ≤ 0.01; ns—non-significant). The error bars represent the standard deviation of the mean. (**b**) Relative mRNA expression levels of p53 target genes in SAL1788 patient cells treated with DMSO or with 5 µM ALRN-6924 for 24 h (*n* = 2, a representative plot from one experiment is shown). (**c**) Flow cytometric analysis of the cell cycle distribution in primary AML cells from two patients (SAL1788 and SYT50) treated with DMSO or with 5 µM ALRN-6924 for 24 h (*n* = 2 for SAL1788, a representative plot is shown; *n* = 1 for SYT50).

## Data Availability

The data presented in this study are available on request from the corresponding author.

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
