# Peer review of "RNAi-Mediated Screen of Primary AML Cells Nominates MDM4 as a Therapeutic Target in NK-AML with DNMT3A Mutations"

_cells, 2022, doi:10.3390/cells11050854_

Round 1

Reviewer 1 Report

Sidorova et al. performed a focused RNAi screen in a panel of 30 primary AML samples, carrying a DNMT3A R882 mutation, and identified MDM4 as a gene essential for proliferation of primary DNMT3AR882X AML. Moreover, they found that the MDM2/MDM4 inhibitor ALRN-6924 impairs growth of DNMT3AR882X primary AML cells in vitro by inducing cell cycle arrest through upregulation of p53 target genes.

Although it is difficult to perform a siRNA screen using shRNAs and primary AML the authors have succeeded in this and identified MDM4 as a target for depletion of primary AML with a DNMT3A mutation. I have several concerns (see below).  

  • The first part of the study, the correction of the mutation in the cell line SET2 is not adding anything to the study and its conclusions. This part should be removed or should be performed in more cell lines and primary AML cells.
  • In the abstract is stated that 30 AML samples are used. Are these all with a DNMT3A mutation and a NPM1 mutation? And no other mutations?  
  • The screen was intended to identify genes that upon targeting deplete primary AML cells that proliferate. Primary AML cells have limited proliferation capacity in culture to my knowledge (some more than others), and the authors have a range of level of cell viability between 1%-55% at 24 hr. The viability data of cells at 20 days (the time point of the end of the screen) should be shown. It seems like more than 60% of the samples (n=10) have still more than 25% of viability after 24 hours. How many % of the samples is still alive after 20 days? Is there a correlation between primary AML samples that have depletion of the positive controls in the screen and proliferative potential and viability of the samples, and the library coverage? All these features should be depicted since it looks like there are almost no cells left after 20 days in most primary AML samples.
  • Supplemental Figure 2A: Could be depicted in the paper itself. Based on these data one might select KMT2D, TRERF1 and MDM4. For TCF19 and TSPYL1 there is no knockdown although there is downregulation of proliferation (or inhibition of apoptosis).
  • Figure 2B could be moved to the supplementary, and Figure 2C could be removed completely.
  • Figure 2D: it looks like these two samples have no loss of viability over time as the scrambled cells do proliferate a little bit? Are these samples the same as the ones from the screening approach or different ones? Although it is difficult to perform triplicate independent experiments with primary AML cells, the experiment in 2D is n=1. This is not sufficient. The authors should have at least performed this experiment in duplicate.
  • Supplemental Figure 3A (but shorter) and 3B could be depicted in the paper and not in the supplementals.
  • It would be interesting to see whether MDM4 expression is different in AML patients with or without DNMT3A mutations in the TCGA dataset, and in patients below and above 60 years of age.
  • Is the inhibitor ALRN-6924 targeting MDM4? Did the authors look at upregulation of MDM4 after treatment? Both should be shown. Like it was already shown by Carvajal et al., ALRN-6924 is efficiently inducing a cell cycle block and inhibiting proliferation in primary AML. However, is it also inducing apoptosis? This experiment should be performed.

Minor comment:

There are several errors starting with “cite Shady’s paper?” in the method section and “error! Reference source not found” several times.

Reviewer 2 Report

The manuscript is very well written. The work is very methodical  and well planned, the results are clearly described and well graphically presented. 

The authors identified MDM4 as a gene essential for DNMT3A-mutation driven AML cells The work is very elegant and well planned, the results are clearly described and well graphically presented. As the authors show that inhibition of MDM4 may serve as a new targeted therapy in AML, it can be expected that the work will be cited multiple times in the literature.

There are only minor revision required associated with reference sources of primers, since some of them were not related to the text (e.g. page 4, line 149, 151, 182, 197)

Round 2

Reviewer 1 Report

The authors have extensively and thoroughly discussed my comments and answered my questions in a satisfied manner, although in too many words.